# The Effect of Strict Lockdown on Omicron SARS-CoV-2 Variant Transmission in Shanghai

**DOI:** 10.3390/vaccines10091392

**Published:** 2022-08-25

**Authors:** Haibo Yang, Hao Nie, Dewei Zhou, Yujia Wang, Wei Zuo

**Affiliations:** 1Shanghai East Hospital, Tongji University School of Medicine, Shanghai 200092, China; 2State Key Laboratory of Respiratory Diseases, Guangzhou Institute of Respiratory Disease, Guangzhou Medical University, Guangzhou 510120, China

**Keywords:** Omicron variant, SARS-CoV-2, stay-at-home, virus transmission

## Abstract

Omicron, the current SARS-CoV-2 variant of concern, is much more contagious than other previous variants. Whether strict lockdown could effectively curb the transmission of Omicron is largely unknown. In this retrospective study, we compared the strictness of government lockdown policies in Shanghai and other countries. Based on the daily Omicron case number from 1 March 2022 to 30 April 2022, the effective reproductive numbers in this Shanghai Omicron wave were calculated to confirm the impact of strict lockdown on Omicron transmission. Pearson correlation was conducted to illustrate the determining factor of strict lockdown outcomes in the 16 different districts of Shanghai. After a very strict citywide lockdown since April 1st, the average daily effective reproductive number reduced significantly, indicating that strict lockdown could slow down the spreading of Omicron. Omicron control is more challenging in districts with higher population mobility and lockdown is more likely to decrease the number of asymptomatic carriers than the symptomatic cases. All these findings indicate that the strict lockdown could curb the transmission of Omicron effectively, especially for the asymptomatic spread, and suggest that differentiated COVID-19 prevention and control measures should be adopted according to the population density and demographic composition of each community.

## 1. Introduction

Since December 2019, the COVID-19 outbreak suddenly and rapidly spread all over the world [1]. As there were no available vaccines and drugs at first, all the countries relied on non-pharmaceutical interventions (NPIs) and applied the lockdown strategy in 2020 as a critical prevention and control measure [2]. Even though vaccines from Pfizer, Moderna, Sinopharm, and other companies were later listed by WHO for emergency use, the emergence of new variants of concern (VOC) such as Delta drove a number of SARS-CoV-2 waves, and lockdowns were still implemented by the majority of governments from time to time [3].

On 2 November 2021, the novel SARS-CoV-2 VOC Omicron was first collected in South Africa and then spread rapidly [4]. It caused abrupt epidemic outbreaks across South Africa, then Europe, and eventually the rest of the world by outcompeting the Delta VOC, which accounted for at least 90% of genomes sequenced globally in October 2021 [4]. This change suggests a strong selective advantage of Omicron as proven by further mutational profile study [5]. In detail, the significant number of mutations in the SARS-CoV-2 receptor-binding domain (RBD) renders the Omicron variant a higher affinity for human angiotensin-converting enzyme 2 than the Delta variant. These changes render the Omicron variant a short mean serial interval of 3 days and an assumed R0 as high as 10 [6,7]. Another important observation in the Omicron epidemic is the reduced odds of hospital admission for patients, which then becomes an implication for the relaxation of public health and social measures (PHSM) as chosen by most countries [8]. However, the epidemiology study of the Hong Kong Omicron wave in early 2022 revealed that the intrinsic severity of Omicron may not be much lower than the ancestral strains [9]. Furthermore, Omicron displays key mutations associated with immune escape (K417N, E484A, T478K in the RBD), forcing researchers to develop novel vaccines for Omicron as the vaccines previously administered to the public are not ideal anymore [10]. Thus, lockdown measures may still be needed facing the Omicron wave, but its effect remains under-explored in the current literature.

In this article, we compared the strictness of the Shanghai lockdown policy with that of other countries. We further evaluated the effect of strict lockdown on Omicron spread in Shanghai since 1 April 2022 and identified critical factors to make the lockdown strategy work optimally. The asymptomatic rate in Shanghai was also analyzed. We expect the experience of the Shanghai Omicron epidemic could provide valuable information to other countries encountering highly transmissible SARS-CoV-2 strains in the future. We would like to suggest authorities formulate differentiated lockdown strategies according to the population density and demographic composition of each community, in which way COVID-19 waves can be controlled effectively with minimal social, economic, and psychological costs.

## 2. Materials and Methods

### 2.1. Data Resource

The daily case number in this Shanghai Omicron wave from 1 March 2022 to 30 April 2022 was retrieved from Shanghai Municipal Health Commission Database. Subway ridership dataset of Shanghai was shared by Chinese Software Developer Network (CSDN).

#### 2.1.1. Timeline: Changes in the PHSM in Shanghai

Stage I: Normal COVID-19 Prevention and Control Measures Stage from March 1st to March 12th when minimum requirements were imposed on majority of the citizens; Stage II: Precise Epidemic Control Stage from March 13th to March 31st where only residents of certain communities at high risk were quarantined; Stage III: Citywide Lockdown Stage since April 1st when the whole city is shut down until gradually opened.

#### 2.1.2. Stringency Index of Government Response

To quantify the strictness of government policies, we followed the Oxford Coronavirus Government Response Tracker project [11] which originally considered nine metrics including: school closing, workplace closing, cancel public events, restrictions on gatherings, closed public transport, stay-at-home requirements, restrictions on internal movement, international travel controls, and public information campaigns. In our research, public information campaign metrics were excluded as the difficulty of accessing information from non-English speaking countries such as South Korea and Japan. A final stringency index was then calculated from the rest eight metrics. Stringency indices of Shanghai at three PHSM stages were calculated separately to visualize the change in government response, and the raw data are shared in Appendix A. For the United States, United Kingdom, German, France, and other countries, their strictest COVID-19 prevention and control policies since 2020 were input and processed in the same way. Their stringency indices were compared with Shanghai Stage III stringency index to assess government responses during lockdown.

#### 2.1.3. The Trend of the Shanghai Omicron Wave

The raw data of daily new infections during the three PHSM stages was smoothed following a 7-day averaged, 4th-order polynomial method. The asymptomatic rate was calculated in the following Equation (1) and smoothed:(1)$$Asymptomatic\;rate=\left(asymptomatic\;cases/daily\;new\;infections\right)*100\%$$

The trend of the Omicron wave in each district was plotted in the same way.

### 2.2. Strict Lockdown Outcome Assessment

#### 2.2.1. Effective Reproductive Number Rt

Estimation of the effective reproductive number Rt is a reliable and common way to evaluate changes in disease transmission over time. It has been widely used in the COVID-19 pandemic to help policymakers and public health officials to assess the effectiveness of interventions [12]. Based on the daily case number at the three PHSM stages, we implemented a time-dependent method for Rt calculation using a previously reported gamma distribution of the Omicron variant [13]. The real data were confirmed to fit the model.

#### 2.2.2. Effective Interval of Lockdown and Its Correlation with Other Factors

In order to better assess the effect of lockdown, we defined the leading time from the lockdown starting date to the daily Omicron case peaking date as the effective interval (EI) of lockdown. We further evaluated the impact of infected cases and population mobility on EI. In this correlation analysis, two algorithms, infection index, and active infection index were involved:(2)$$Infection\;index=\ln\left(daily\;new\;infections\;per\;one\;million\;people\right)$$

The infection index only considers the number of daily Omicron cases on the day before lockdown.
(3)$$Active\;infection\;index=\ln\left(daily\;new\;infections\;per\;one\;million\;people\right)\times \ln daily\;subway\;ridership$$

The active infection index considers both the number of daily Omicron cases on the day before lockdown and population mobility. In Equation (3) the daily subway ridership was used as an indicator of the population mobility [14].

The correlation analysis between these two indices with the EI of 16 districts in Shanghai was examined by two-tailed Pearson correlation in GraphPad Prism 9.3.1.

## 3. Results

### 3.1. Stringency Index of Shanghai Lockdown

Shanghai is one of the largest cities in the world with a population of 25 million. On 1 March 2022, Omicron variant BA.2 hit Shanghai and the case number increased rapidly in the following weeks. The Shanghai government decided to implement a “static management” lockdown in the eastern half of the city on March 28th and then in the whole city on April 1st, to curb Omicron from rapid spreading. During the lockdown time, all schools and workplaces were closed except for necessary health care services including hospitals and COVID-19 testing providers. Public transport was closed, and a “stay-at-home” order was given to almost all residents in the city except those working on Omicron prevention and control. All public events were canceled, and gatherings were restricted. On June 1st, Shanghai started to open up gradually, tentatively, and cautiously.

The calculated stringency index of the Shanghai lockdown is 97, which is only slightly lower than the 100 of the India lockdown, but higher than all other countries (Figure 1). Shanghai lockdown gets 100 marks in seven metrics and 75 marks in international travel controls. By then, India totally closed its international borders and only international flights with special permission to conduct cargo operations were allowed during their lockdown period [15]. Instead, Shanghai has been adopting circuit breaker arrangements for international passenger flights since the start of the COVID-19 pandemic. In detail, designated airline companies are allowed to operate one international flight between Shanghai and another city every week. The passengers will be quarantined for two weeks, and the flight will be paused for a certain period according to the number of COVID-19 cases among these passengers if there are any. Thus, Shanghai already imposed as many restrictions as possible during the lockdown, which is one of the strictest worldwide.

### 3.2. Omicron Transmission before and after the Lockdown in Shanghai

As aforementioned in the Methods section, we classified three PHSM stages. The daily stringency index during Stage I was merely nine as only individuals with a travel history of the medium- or high-risk regions were monitored. As Shanghai started to control certain communities at high risk in a “2+12” manner in Stage II, the daily stringency index became 45. After the “static management” in the eastern half of Shanghai on March 28th, the daily stringency index further increased to 66. Implementation of lockdown policy in Stage III finally reached a daily stringency index of 97 (Figure 2, green squares).

Next, we calculated Rt, the effective reproductive number indicating changes in disease transmission over time, of the three stages to analyze the Omicron transmission under different PHSM policies. The change of Rt during the Omicron wave was shown in Appendix A. Daily averaged Rt of Stage I and Stage II are 1.76 (95% Cl: 1.44 to 2.09) and 1.79 (95% Cl: 1.7 to 1.89), respectively. These Rt are lower than the average Rt of 3.4 in the Omicron epidemics of South Africa, the UK, the Netherlands, and India [16]. After the citywide lockdown of Shanghai (Stage III), the Rt decreased significantly to 1.04 (95% Cl: 1.03 to 1.06), demonstrating that lockdown can effectively prevent the highly transmissible Omicron from spurting.

Accordingly, after imposing the lockdown on April 1st, the daily Omicron cases in Shanghai peaked on April 12th, meaning an EI of 11 days (Figure 2). This is significantly shorter than the EI of 20 days during the lockdown measures of Wuhan, China in 2020 facing the original SARS-CoV-2 strain [17], probably due to a stricter lockdown policy in Shanghai than that in Wuhan.

### 3.3. Control of Omicron in 16 Districts of Shanghai after Lockdown

Shanghai city has 16 districts, including both urban regions with high population density and also rural areas on its outskirt. We further compared the trend of Omicron waves in all the 16 districts of Shanghai (Figure 3). Interestingly, it was noticed that the 16 different districts have distinct EI days ranging from 6 days to 20 days (mean ± SD: 10.94 ± 4.09). Double peaks were observed in a few districts, including Hongkou, Yangpu, and Baoshan, as predicted by some previous lockdown modeling studies [18]. These two smaller peaks might be inevitable to relieve the burden on the healthcare system.

We tried to interpret the significant difference in EI in the 16 districts under the same lockdown policy. As more infections will put a bigger population at risk of Omicron, we analyzed the impact of daily new infections before lockdown on EI. Infection index, an algorithm only considering daily new infections, was introduced and its calculation was given in Equation (2). The infection index of each district was then plotted against the corresponding EI as shown in Figure 4A. However, the low correlation (r) value of 0.416 and the insignificant *p*-value of 0.109 indicate that daily new infections before lockdown alone cannot determine the EI. Since only Omicron carriers who are in contact with others can form a transmission chain, we introduced an active infection index, another algorithm that takes both daily new infections and population mobility into account. In Equation (3) of the active infection index, the daily subway ridership in each district of Shanghai was adopted as the indicator of population mobility [14]. The correlation analysis between EI and active infection index in Figure 4B revealed a significant positive correlation (r = 0.5974, *p* = 0.0145), showing both the higher number of infected individuals and the extensive personnel movement before lockdown brings challenges to curbing the Omicron epidemic. This result demonstrates that timely lockdown before the surge of case number is important for rapid control of Omicron transmission, especially for those urban regions with high population mobility.

### 3.4. The Asymptomatic Rate in Shanghai Omicron Wave

Asymptomatic spread is a characteristic feature of Omicron [19]. In this Shanghai Omicron wave, the asymptomatic rate fluctuated above 80% (Figure 5A), which might be attributed to factors including the traits of Omicron, the vaccination rate, and early detection of infection cases under mass testing. According to Shanghai Municipal Center for Disease Control and Prevention, people aged 60 or under who are in good health have accounted for 84.5% of cases, pushing up the asymptomatic rate. In addition, more than 88% of Shanghai residents are fully vaccinated. Even though the immunoevasive property of Omicron brings difficulty for vaccines to achieve full protection against infection and transmission, their effectiveness against symptomatic diseases, hospitalization, and mortality are also precious as pointed out by COVID-19 vaccine weekly surveillance reports of the United Kingdom Health Security Agency [20].

In order to understand the effect of lockdown on the asymptomatic transmission of Omicron, we analyzed the asymptomatic rate before and after lockdown (Figure 5B). The data showed that during Stage I and Stage II, the median asymptomatic rate was 94.67% and 96.84%, respectively. After lockdown, the median asymptomatic rate of Stage III dropped to 90.11%. It is unclear that why there is a significant decrease in asymptomatic transmission after the lockdown. Regardless of the possible change in PCR testing policy, one explanation could be that asymptomatic cases are more common in young and middle-aged individuals [21]. Young people are more active in social contact before lockdown; therefore, their infection risk is sensitive to lockdown. In contrast, older-age residents tend to develop symptoms because of their weakened immune systems. Their activity is more limited in family or close neighborhoods (neighborhoods sharing kitchens and/or toilets). Accordingly, the city lockdown has little effect on the Omicron transmission in family or close neighborhood clusters, which explains the change in the asymptomatic/symptomatic ratio.

## 4. Discussion

In the Shanghai Omicron wave, the municipal government implemented the strictest lockdown policy. These measures effectively stopped the spreading of Omicron, especially its asymptomatic transmission. Through correlation analysis, we found that timely lockdown before the surge of case numbers is critical for putting the Omicron epidemic under control in urban regions. Previous work also utilized the same Oxford Coronavirus Government Response Tracker project to evaluate the association between physical distancing interventions and the incidence of COVID-19 in 149 countries [22]. Among the nice metrics, their primary interventions of interest were those aimed at physical distancing, including the closure of schools and workplaces, restrictions on mass gatherings, public transport closure, stay-at-home regulations, and restrictions on movements within a country [22]. The association between the sequence of interventions and the change in the incidence of COVID-19 was analyzed. They found a greater decrease in the incidence of COVID-19 was associated with the earlier implementation of lockdown rather than later implementation [22]. This is consistent with our correlation analysis implying that timely lockdown before the surge of case number is important for rapid control of Omicron transmission.

Some other previous studies investigated the impacts of New Zealand’s graduated, risk-informed national COVID-19 suppression measures in early 2020 on the epidemiology of COVID-19 [23]. They did a descriptive epidemiological study of all laboratory-confirmed and probable cases of COVID-19, which makes their results much more reliable and convincing. A feature of this New Zealand’s COVID-19 pandemic is the low proportion of asymptomatic infection compared with other countries despite widespread testing. The low level of community transmission was believed to contribute to this. This supports our hypothesis that impeded community transmission after the lockdown in Shanghai reduced asymptomatic transmission between young and middle-aged residents, leading to a lower asymptomatic rate at Stage III.

In previous studies, Patricio et. al. conducted an empirical analysis of the impact of lockdown on SARS-CoV-2 transmission and death toll for a panel of 152 countries [24]. This study covered the time period from the start of the pandemic through 31 December 2020, about two previously circulating VOCs, Alpha and Beta [25]. They provided evidence that restrictions indeed had a significant effect in the first weeks after the policy introduction. The effect of NPIs on the reproductive number Rt peaked at about 10 days and disappeared at about 20. However, the authors emphasized that after 120 (continuous or discontinuous) days of strict lockdown, neither the SARS-CoV-2 spread nor daily deaths per million can be reduced by the lockdown. The authors concluded that restrictions played a role early on in the pandemic but had only transient effects that were difficult to replicate going forward. Thus, more attention should be paid to the social, economic, and psychological costs at the late stage of strict lockdown to adjust PHSM on time [26].

Due to privacy concerns, COVID-19 data acquired and released by governments and healthcare authorities are mostly only available at a relatively coarse level, which is usually limited to the nationwide level instead of down to local areas [27,28]. Of note, Gao et al. presented a precious community-level study about ancestral SARS-CoV-2 strain transmission and policy interventions in Wuhan, China [29]. The authors found that the viral spread in local communities was irrelevant to the socioeconomic position and the built environment of a community but was related to its demographic composition. Even though the elderly were more vulnerable to virus infection and had a higher mortality rate, the mobility of the young population was more associated with viral transmission in local communities. In our current work, we also attributed the significant decrease in asymptomatic transmission after lockdown during the age of Omicron to the young and middle-aged individuals whose mobility is limited more severely by lockdown measures. Both these two studies suggest that differentiated lockdown measures should be carried out on young people and the elderly, as the elderly are less likely to be a reservoir of the chain of infection (until they are diagnosed) and the limited food, nutrition, and medical supplies can easily threaten their life and health.

### Strengths and Limitations

The epidemiological data analyzed in this study were collected through numerous thoughtful PCR tests during the 2-month lockdown. Considering that majority of the countries have been adopting loose PHSM, this Omicron outbreak in Shanghai provides a precious chance to accurately evaluate the effectiveness of strict lockdown measures on curbing the spreading of the highly contagious Omicron variant. However, the estimation of population mobility in this study was limited to subway ridership, as other public transportation and personal vehicle information are unavailable.

## 5. Conclusions

After its continuous evolution in the past two years, the new strains of SARS-CoV-2 keep presenting new features that bring challenges to COVID-19 prevention and control. This study confirmed that strict lockdown measures are capable of effectively curbing the transmission of Omicron, the current highly contagious VOC. The success of lockdown measures in Shanghai can be largely attributed to their potency in cutting the asymptomatic spread, which can trigger the COVID-19 explosion unexpectedly. Based on our studies, we suggest that differentiated lockdown strategies should be formulated for each community according to its population density and demographic composition in order to reduce the social, economic, and psychological costs of city-wide lockdown.

## Figures and Tables

**Figure 1 vaccines-10-01392-f001:**
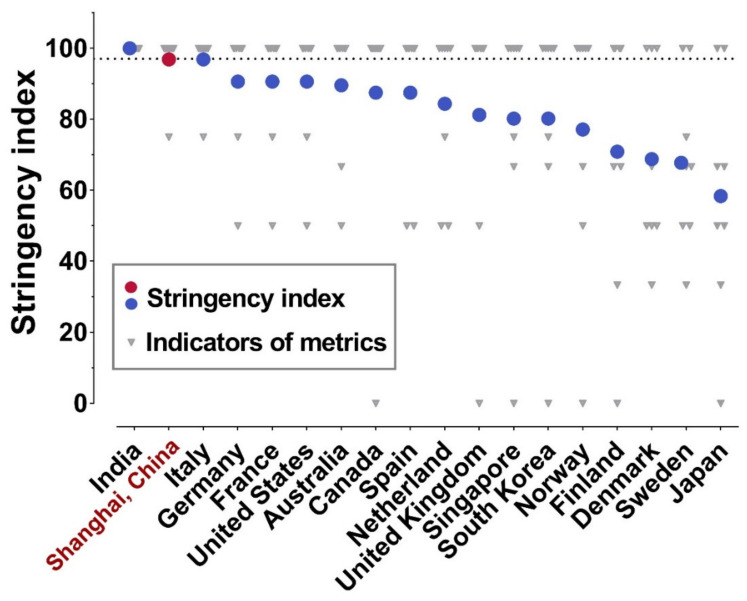
Stringency indices of lockdown policies implemented by different governments. Based on the Oxford Coronavirus Government Response Tracker project, the lockdown policies of different governments were assessed considering eight metrics (plotted as grey triangles). The stringency indices of lockdown policies were then derived from the eight metrics and presented as red/blue dots.

**Figure 2 vaccines-10-01392-f002:**
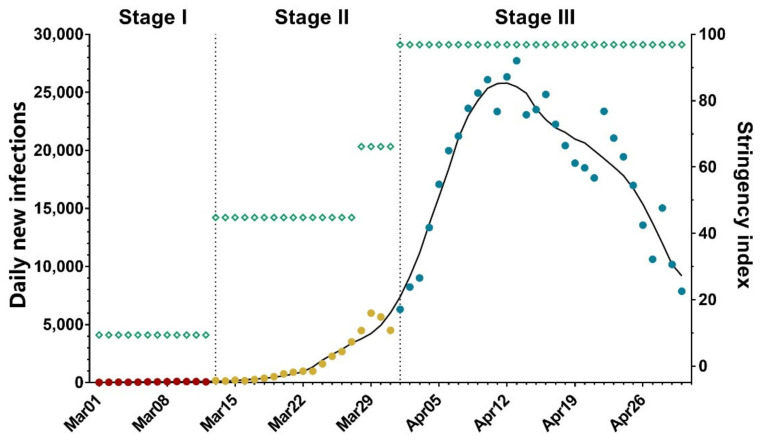
New daily infections in Shanghai with corresponding stringency index. The new daily infections (red, yellow, and blue dots indicate numbers during the Public Health and Social Measures Stage I, II, and III, respectively), with corresponding stringency index (green squares) in Shanghai from 1 March 2022 to 30 April 2022. Dashed vertical lines indicate the beginning date of Stage II and III (March 13th and April 1st, respectively). The raw data were smoothed following a 7-day averaged, 4th-order polynomial method, and the trend was plotted as black curve for daily new infections.

**Figure 3 vaccines-10-01392-f003:**
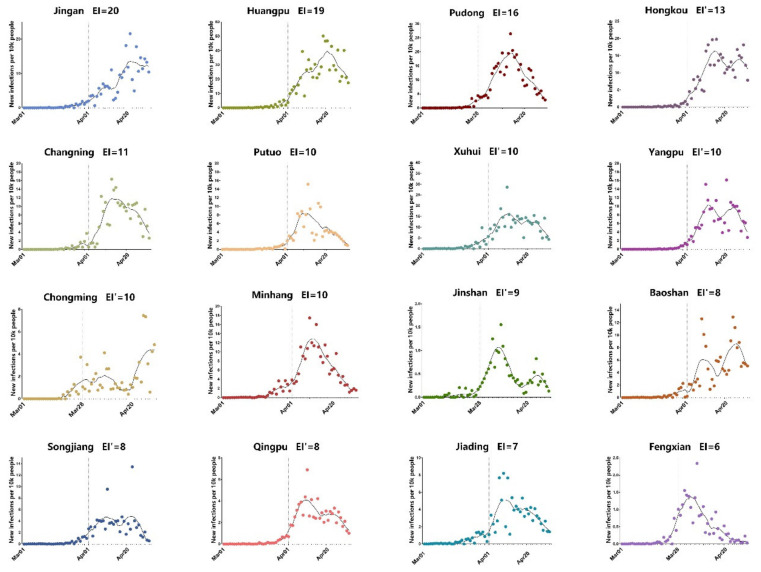
Daily new infections in 16 districts of Shanghai. The daily new infections from 1 March 2022 to 30 April 2022 in 16 districts of Shanghai were plotted as dots. The raw data were smoothed following a 7-day averaged, 4th-order polynomial method, and the trends were plotted as black curves. Dashed vertical lines indicate the lockdown date of each district. EI, the leading time from the lockdown starting date to the daily omicron cases peaking date, was labeled for each district. For districts with double peaks, EI was identified based on the first peak and differentiated as EI’.

**Figure 4 vaccines-10-01392-f004:**
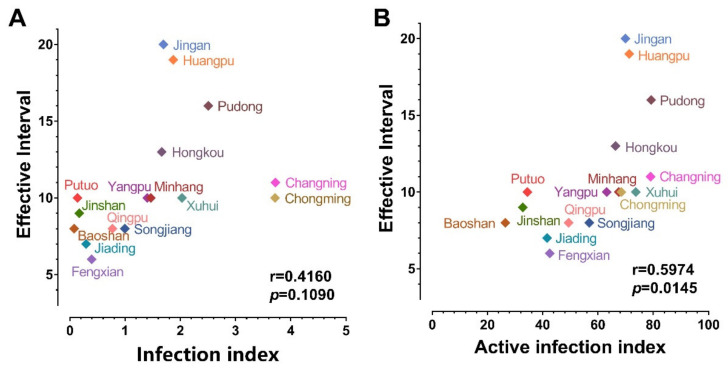
Correlation analysis between EI and the calculated infection index/active infection index. EI is the leading time from the lockdown starting date to the daily omicron cases peaking date in each district. The calculated infection index is an algorithm considering only the daily new infections number in each district. The calculated active infection index is an algorithm taking both the number of infected individuals and the population mobility in each district into account. Data are examined by two-tailed Pearson correlation in GraphPad Prism 9.3.1. (**A**) The correlation between EI and the calculated infection index. (**B**) The correlation between EI and the calculated active infection index.

**Figure 5 vaccines-10-01392-f005:**
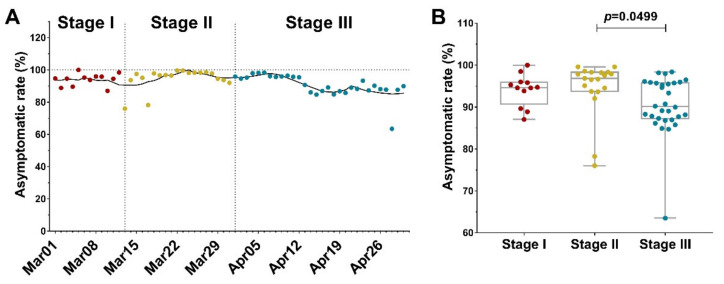
Asymptomatic rate in Shanghai Omicron wave. Asymptomatic rate is the ratio of asymptomatic cases to daily new infections. (**A**) Asymptomatic rate (red, yellow, and blue dots for the PHSM Stage I, II, and III, respectively), in Shanghai Omicron wave. Dashed vertical lines indicate the beginning date of Stage II and III (March 13 and April 1, respectively). The data were smoothed following a 7-day averaged, 4th-order polynomial method, and the trend was plotted as black curve. (**B**) The comparison of asymptomatic rate during the three PHSM Stages.

## Data Availability

All the raw data are available in the Appendix A.

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
