# Peer review of "The Effect of Strict Lockdown on Omicron SARS-CoV-2 Variant Transmission in Shanghai"

_vaccines, 2022, doi:10.3390/vaccines10091392_

Round 1

Reviewer 1 Report

The authors present an interesting retrospective study on the effectiveness of government policies to counter the spread of the Omicron variant of SARS-CoV2 in Shanghai in March-April 2022.

Overall, the article appears clear and well presented.

The references are all related to recently published papers (within the last 5 years), as these are articles related to COVID-19, a pandemic that exploded in late 2019.

In a deontologically correct manner, no self-citations have been included.

The study design appears technically correct and appropriate to answer the researchers' questions. The entire methodology is precisely and comprehensively explained in the 'Materials and Methods' section.

A wealth of documents included in the supplementary material (including raw data) helps the reader to understand the methodology used in detail.

The illustrated figures are very easy to interpret and adequately presented within the text. 

The conclusions are consistent with the evidence presented, although I would suggest that they are illustrated in a dedicated paragraph (a paragraph entitled 'Conclusions' is missing, the conclusions being illustrated within the 'Discussion' paragraph).

To sum up, I find the article to be of high scientific and technical quality and I do not see the need to make any substantial changes, other than to make the conclusions more directly explicit by introducing a dedicated paragraph.

Another thing I would suggest is to briefly discuss the implications of the results of this study on policies to contain the spread of the COVID-19 pandemic globally. That is, to answer the question: "how can the results of this study have an impact on global lockdown policies?" (this could be included in the 'Conclusions' section).

Reviewer 2 Report

This is an interesting study investigating the effect of strict lockdown on Omicron SARS-CoV-2 variant transmission in Shanghai. The authors concluded that the strict lockdown could curb the transmission of Omicron effectively, especially for the asymptomatic spread.

I have a few minor comments:

In the Abstract section (L.21): Avoid starting the sentence with "And", You can delete it.

Keywords: Use a synonym of "Lockdown" because the word Lockdown appears in the Title.

In the Introduction, add a reference for the following sentence: "On November 2nd 2021, the novel SARS-CoV-2 VOC Omicron was first collected in South Africa and then spread rapidly."

The authors wrote: "Thus, lockdown measures may still be needed facing Omicron wave but its effect remains largely unknown." Are you sure that your study was the first to evaluate the effect of strict lockdown on Omicron SARS-CoV-2 variant transmission in Shanghai?

What about Delta variant and strict lockdown? Please add in the Introduction

In the Discussion, L255-262: Please add a reference

The Discussion section should be expanded by adding some studies evaluating the effects of strict lockdown (to mitigate other variants in China and other countries).

Surprisingly, Limitation section is lacking and also a Conclusion section is also absent. The authors must add Limitations and a conclusion section.

Round 2

Reviewer 2 Report

The authors responded to all my comments. I suggest the publication of the article.